# Metal–peptide rings form highly entangled topologically inequivalent frameworks with the same ring- and crossing-numbers

Tomohisa Sawada [1], Ami Saito [1], Kenki Tamiya [1], Koya Shimokawa [2], Yutaro Hisada [1] & Makoto Fujita[1]

With increasing ring-crossing number ($c$), knot theory predicts an exponential increase in the number of topologically different links of these interlocking structures, even for structures with the same ring number ($n$) and $c$. Here, we report the selective construction of two topologies of 12-crossing peptide [4]catenanes ($n = 4$, $c = 12$) from metal ions and pyridine-appended tripeptide ligands. Two of the 100 possible topologies for this structure are selectively created from related ligands in which only the tripeptide sequence is changed: one catenane has a $T_2$-tetrahedral link and the other a three-crossed tetrahedral link. Crystallographic studies illustrate that a conformational difference in only one of the three peptide residues in the ligand causes the change in the structure of the final tetrahedral link. Our results thus reveal that peptide-based folding and assembly can be used for the facile bottom-up construction of 3D molecular objects containing polyhedral links.

[1] Department of Applied Chemistry, School of Engineering, The University of Tokyo, 7-3-1 Hongo, Bunkyo-ku, Tokyo 113-8656, Japan. [2] Department of Mathematics, Saitama University, 255 Shimo-Okubo, Sakuraku, Saitama 338-8570, Japan. Correspondence and requests for materials should be addressed to T.S. (email: tsawada@appchem.t.u-tokyo.ac.jp) or to M.F. (email: mfujita@appchem.t.u-tokyo.ac.jp)

Knot theory deals with the topological links of multiple rings[1]. According to this theory, increasing ring ($n$) and ring-crossing ($c$) numbers give rise to a greater number of topologically different links[2,3]. However, even for structures with the same $n$ and $c$ numbers, the theory predicts a large number of topologically different links when $c - n$ is >7 (Fig. 1a). Recently, we reported the folding and assembly of 12-crossing peptide [4] catenane 2 ($n = 4$, $c = 12$; this framework is hereafter abbreviated as [4]$_{12}$-catenane), which have the largest crossing number among all synthetic well-defined interlocking molecules[4–15] to date, from tripeptide ligand 1 and Ag(I) ions (Fig. 2a)[16]. The formation of this highly entangled structure is driven by the intrinsic folding nature of the PGP sequence in ligand 1 (P: L-proline, G: glycine) into an $\Omega$-shaped loop conformation, and by Ag(I) coordination to ligand 1, which induces both self-assembly and peptide folding. In case of >11 crossings, the existence of very large number of topologies prevents us from assigning the topology by standard link tables as seen in previous topological molecules[4–15], accordingly we choose to use the polyhedral link description here: the topology of 2 is classified as a $T_2$-tetrahedral link in knot theory[17], and is, surprisingly, one of 100 possible topological links of [4]$_{12}$-catenanes[2,3].

Here, we report the folding and assembly of another [4]$_{12}$-catenane with a different topological link. New tripeptide ligand 3, which contains a TPP sequence (T: L-threonine), is found to fold and assemble into [4]$_{12}$-catenane 4 (Fig. 2b, c), with a topology classified as a three-crossed tetrahedral link[18] or the mathematically equivalent cuboctahedral link[19] (Fig. 1b). The folding of the TPP sequence into a polyproline II helix ($P_{II}$-helix) conformation is key to the generation of this alternative topological link. We thus suggest that concerted folding and assembly may be a rational strategy for the generation of highly complex polyhedral link structures with $c > 10$.

## Results

**Synthesis and characterisation of [4]$_{12}$-catenane 4.** Tripeptide ligand 3 was synthesised by solution-phase peptide synthesis (Supplementary Methods), and then mixed with AgOTf (1 equiv., 50 mM) in $CD_3NO_2$ (0.5 mL) at ambient temperature for 3 days to achieve the self-assembly of [4]$_{12}$-catenane 4. $^1H$ NMR measurements showed that ligand 3 exists as a mixture of conformers, which converge into a single conformer after coordination to Ag (I) (Fig. 3a–c and see also Supplementary Fig. 11a, b). The highly dispersed aromatic and amide proton signals observed between 6.9 and 10.7 ppm are indicative of the formation of a highly entangled structure and have previously been observed in peptide-based catenanes[16,20]. Diffusion-ordered spectroscopy (DOSY) measurements also support the formation of a single product (Supplementary Fig. 10).

The $Ag_{12}(3)_{12}$ composition of the self-assembled product was clearly confirmed by electrospray ionisation-time of flight (ESI-TOF) mass spectrometry supported by ion mobility separation[21], which allowed the multiply charged overlapping ion peaks at around $m/z = 2105$ to be fully separated on the basis of their mobility ($K_0^{-1}$) (Fig. 3d). The high-resolution peaks corresponding to $[Ag_{12}(3)_{12}(OTf)_8]^{4+}$ were clearly observed at $K_0^{-1} = 1$ (calcd. 2105.30, found 2105.29) after the separation of unavoidable fragmentation peaks, such as $[Ag_6(3)_6(OTf)_4]^{2+}$ ($K_0^{-1} = 1.5$) and $[Ag_3(3)_3(OTf)_2]^+$ ($K_0^{-1} = 2$) (Fig. 3e).

The molecular structure of [4]$_{12}$-catenane 4 was revealed by single crystal X-ray analysis. Single crystals were obtained by slow vapour diffusion of diethyl ether into a $CH_3NO_2$ solution of 4 (initial concentration [4] = 1.7 mM) at ambient temperature. The 2-nm-sized, roughly spherical structure of 4 with an $Ag_{12}(3)_{12}$ composite was confirmed by crystallographic analysis (Supplementary Table 1 and Supplementary Fig. 1a). The molecule is composed of four equivalent $Ag_3(3)_3$ rings arranged in $T$ symmetry with respect to each other (Supplementary Fig. 3). Ligand 3 is unidirectionally arranged in each ring, and the four rings are interlocked in such a way that any two rings are singly interlocked; this results in [4]$_{12}$-catenane 4. Peptide catenane 4 has the same $n$ and $c$ numbers as previously reported catenane 2[16], however the two catenanes are different polyhedral links. This difference is clearly illustrated by the cartoon representation of their metal–peptide backbones in Fig. 2. To the best of our knowledge, this is the first example of the chemical construction of different polyhedral links with the same $n$ and $c$ numbers and the same length ring components.

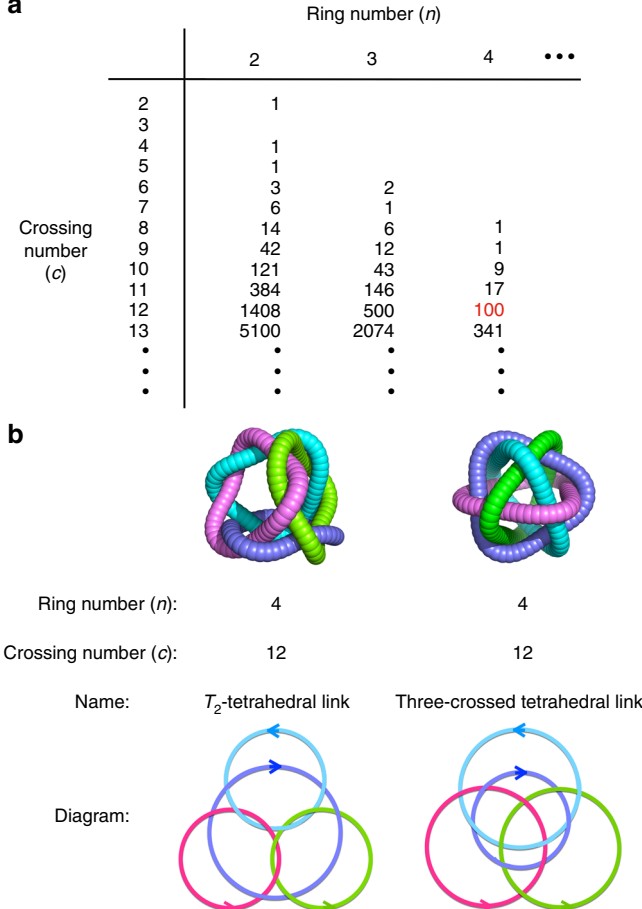

**a**

Ring number ($n$)

| | 2 | 3 | 4 | ••• |
|---|---|---|---|---|
| 2 | 1 | | | |
| 3 | | | | |
| 4 | 1 | | | |
| 5 | 1 | | | |
| 6 | 3 | 2 | | |
| 7 | 6 | 1 | | |
| 8 | 14 | 6 | 1 | |
| 9 | 42 | 12 | 1 | |
| 10 | 121 | 43 | 9 | |
| 11 | 384 | 146 | 17 | |
| 12 | 1408 | 500 | 100 | |
| 13 | 5100 | 2074 | 341 | |

Crossing number ($c$) (row labels at left)

**b**

| | |
|---|---|
| Ring number ($n$): | 4 | 4 |
| Crossing number ($c$): | 12 | 12 |
| Name: | $T_2$-tetrahedral link | Three-crossed tetrahedral link |
| Diagram: | | |

**Fig. 1** Number of topologies and overview of the two [4]$_{12}$-catenane topologies discussed in this work. **a** Number of possible topologies corresponding to ring number ($n$) and crossing number ($c$) in the case of prime, unoriented, alternating links[2]. **b** Name and diagram of two topologies of $(n, c) = (4, 12)$ constructed in this work. Since each of these links has a reduced alternating link diagram[1] of 12 crossings, its crossing number is exactly 12. Each ring has a directional sense, which is defined by the N → C direction within each peptide ligand here

**Topological selection by the tripeptide sequence.** The topological selection in the formation of these [4]$_{12}$-catenanes depends on the choice of amino acid sequence in the ligand. We examined other analogous ligands with XPP peptide sequences, where X is an amino acid residue with an alkyl side chain (Fig. 4, Table 1 and Supplementary Methods). With the APP (A: L-alanine) sequence (ligand 5), selective formation of the three-crossed-type [4]$_{12}$-catenane (6) was clearly confirmed by a crystallographic study

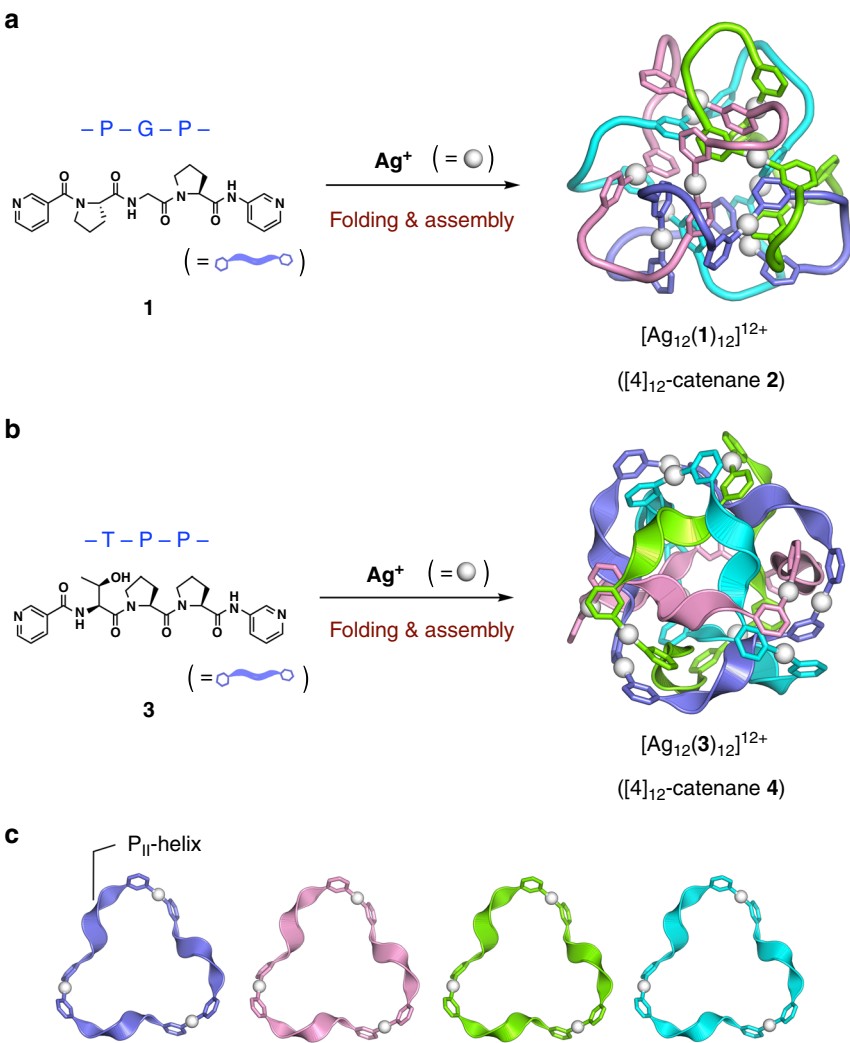

**Fig. 2** Formation of two types of [4]$_{12}$-catenane. **a** Schematic representation of the formation of [4]$_{12}$-catenane **2** by Ag(I) ion-mediated folding and assembly of tripeptide ligand **1**, which was previously reported by our group[16]. **b** New [4]$_{12}$-catenane topology **4**, obtained through the same folding-and-assembly strategy but using tripeptide ligand **3**. **c** Four Ag$_3$(**3**)$_3$ macrocyclic components based on the P$_{II}$-helix conformation extracted from the structure of **4**

(Supplementary Figs. 1b and 4), $^1$H NMR measurements (Supplementary Fig. 11c, d), and mass spectrometry (MS) analysis (Supplementary Fig. 9). The peptide P$_{II}$-helix conformation was revealed to be vital for the formation of the three-crossed unit (Supplementary Fig. 2). In sharp contrast, the IPP (I: L-isoleucine) sequence (ligand **7**) resulted in the $T_2$-type [4]$_{12}$-catenane (**8**) as a single product (Supplementary Figs. 1c, 11e, f, and 12). In this case, formation of the Ω-shaped loop conformation was induced by the *cis*-form of the amide bond of the PP sequence, presumably due to the neighbouring bulky *sec*-butyl side chain (Supplementary Fig. 2). The same behaviour was observed for the VPP (V: L-valine) sequence (ligand **9**) in $^1$H NMR observation: the $T_2$-type [4]$_{12}$-catenane (**10**) was formed (Supplementary Fig. 11g, h). Variable temperature NMR analysis reveals that the topological interconversion between three-crossed type and $T_2$-type [4]$_{12}$-catenane does not occur in solution from 273 to 353 K (Supplementary Figs. 16a and 19). Similarly, the concentration change before and after the complexation also does not affect the topological selectivity (Supplementary Figs. 16b and 17) except for the formation of the stable intermediates at the low concentration in case of **4** (Supplementary Fig. 14). Thus, the topology of the polyhedral link product is solely determined by the choice of the peptide sequence.

**Topological analysis of the two [4]$_{12}$-catenanes**. The topological links of [4]$_{12}$-catenanes **2** and **4** were analysed based on knot theory (see also Supplementary Discussion). The topology of **2** belongs to a doubly twisted tetrahedral link ($T_2$-tetrahedral link)[17], in which four equivalent rings are placed on the four faces of a tetrahedron and entwined by twisting the strands twice ($T_2$ operation) at every edge of the tetrahedron. The crossing number of 12 arises from the $T_2$ operation at six edges ($2 \times 6$) (Fig. 5a, arrow A). Chemically, the $T_2$ twisting at each edge is generated by the entanglement of two Ω-shaped PGP loops (Fig. 6a). In the same manner, the topology of **8** and **10** also belongs to $T_2$-tetrahedral link. In sharp contrast, the topology of **4** is described as a three-crossed tetrahedral link[18], in which three strands are crossed with each other at the four vertices of a tetrahedron ($3 \times 4$), giving the same crossing number of 12 (Fig. 5b, arrow B). Notably, the topology of **4** can also be described as a cuboctahedral link, in which the four rings are singly crossed at the 12 vertices of a cuboctahedron ($1 \times 12$). Figure 5b and c clearly illustrates that the three-crossed tetrahedral link and the cuboctahedral link can be topologically transformed into the other without the need to cut any loops[19]. The topology of **6** is the same as that of **4**.

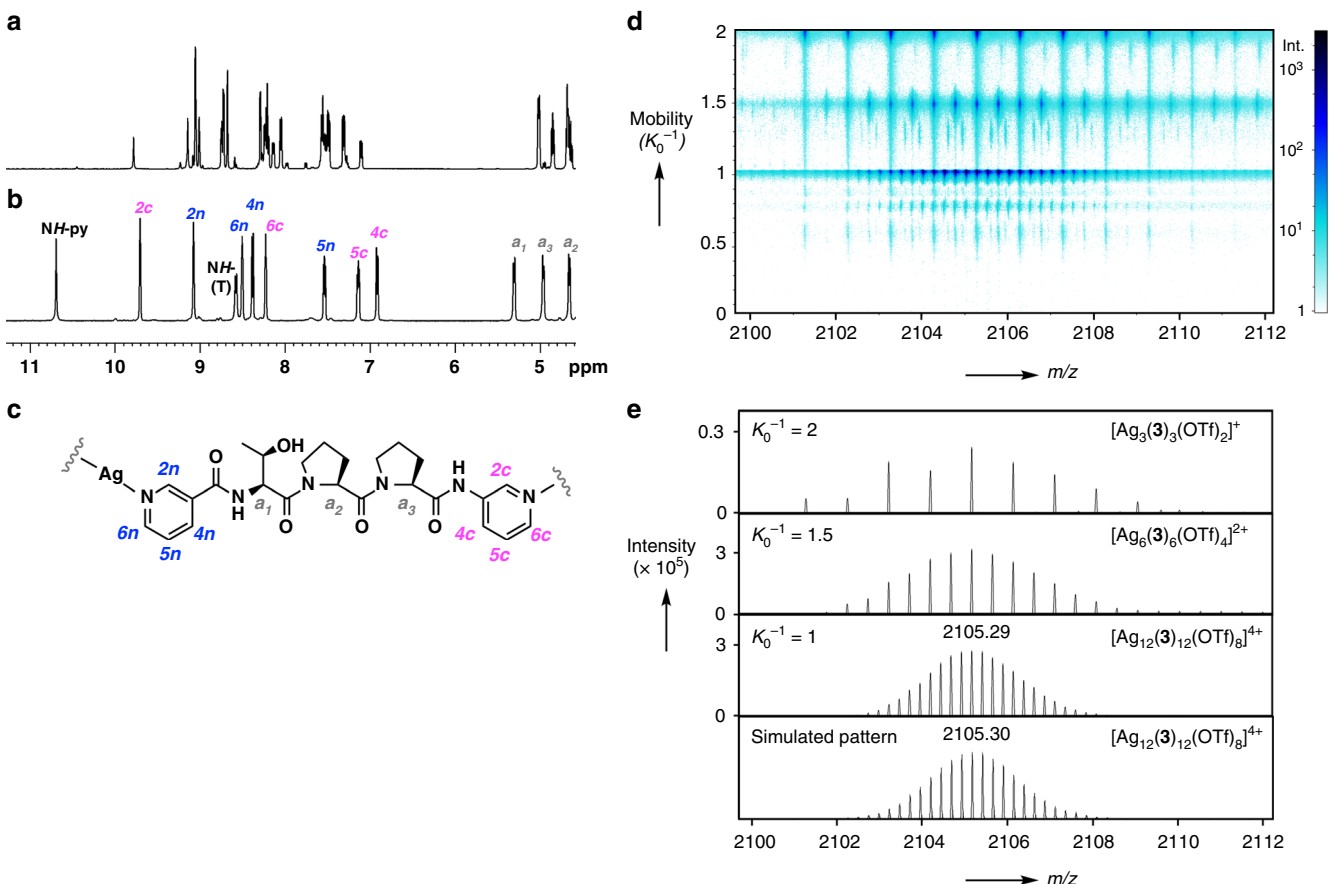

**Fig. 3** $^1$H NMR and ion mobility mass spectrometry data of [4]$_{12}$-catenane **4**. **a** Partial $^1$H NMR spectrum of ligand **3** in CD$_3$NO$_2$. **b** Partial $^1$H NMR spectrum of [4]$_{12}$-catenane **4** in CD$_3$NO$_2$ (300 K, [**3**] = 50 mM). **c** Chemical structure of ligand **3** with proton numbers used for the signal assignments in **b**. **d** Mobility vs. m/z plot obtained from ion mobility mass spectrometry of **4**. **e** Mass spectra at $K_0^{-1} = 2$, $K_0^{-1} = 1.5$, $K_0^{-1} = 1$, and simulated pattern of [Ag$_{12}$(**3**)$_{12}$(OTf)$_8$]$^{4+}$ from the top

**Fig. 4** Examination of ligands **5**, **7**, and **9** for the [4]$_{12}$-catenane formation

**Table 1 Types of [4]$_{12}$-catenanes formed in terms of the first amino acid residue in the tripeptide ligand sequence**

| Ligand | Peptide sequence | Peptide conformation | [4]$_{12}$-Catenane type |
|---|---|---|---|
| 1 | –P–G–P– | Ω-shaped loop | $T_2$-tetrahedral link |
| 3 | –T–P–P– | P$_{II}$-helix | Three-crossed tetrahedral link |
| 5 | –A–P–P– | P$_{II}$-helix | Three-crossed tetrahedral link |
| 7 | –I–P–P– | Ω-shaped loop | $T_2$-tetrahedral link |
| 9 | –V–P–P– | Ω-shaped loop | $T_2$-tetrahedral link |

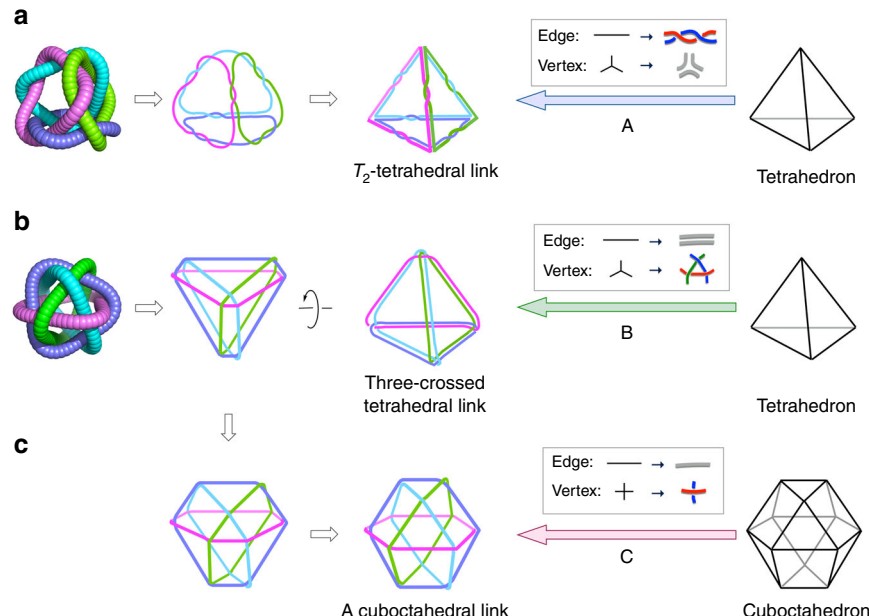

**Fig. 5** Illustrations of the two $[4]_{12}$-catenane topologies. **a** Topology of $[4]_{12}$-catenane **2** constructed from a tetrahedron through method A, in which each edge is converted to $T_2$-double lines and each vertex to three separated lines. **b** Topology of $[4]_{12}$-catenane **4** constructed from a tetrahedron through method B, in which each edge is converted to two separated lines and each vertex to three-crossed lines. **c** Another topological description of **4**, in which it is constructed from a cuboctahedron through method C. Each edge is retained as a single line and each vertex converted to a raised intersection. Note that the direction of each loop in this figure is not shown

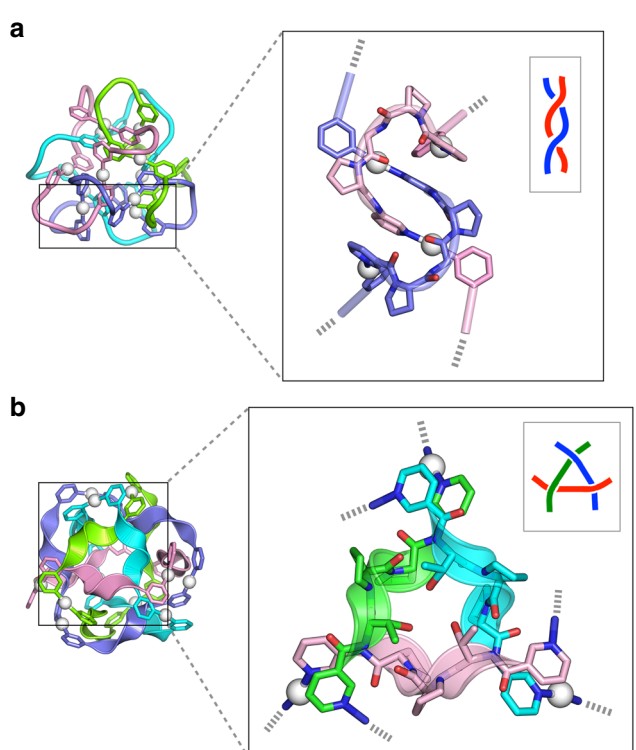

**Fig. 6** Structural origins of the molecular entanglements observed in $[4]_{12}$-catenanes. **a** $T_2$-edge geometry induced by interlocked Ω-shaped loops of **1** extracted from the crystal structure of $[4]_{12}$-catenane **2**[16]. **b** Three-crossed geometry induced by the $P_{II}$-helical conformation of **3** observed in the crystal structure of $[4]_{12}$-catenane **4**. Peptide regions are highlighted as cartoon representations. Hydrogen atoms, counter anions, and solvents have been omitted for clarity

We note that the three-crossed junction at each vertex of the tetrahedron is the key motif that leads to the newly obtained tetrahedral link in **4**. The crystal structure reveals that the $P_{II}$-helix conformation of ligand **3** orients the threonine hydroxy group toward the vertex of the tetrahedron; this forms a hydrogen-bonded cyclic trimer that probably induces and stabilises the three-crossed junction despite the bulky side chain of the T residue, which is similar to the side chains in the I and V residues of ligands **7** and **9** (Fig. 6b and see Supplementary Fig. 5). The chemical stability of the three-crossed $Ag_3(\mathbf{3})_3$ is suggested by the intense $[Ag_3(\mathbf{3})_3]_m$ ($m = 1, 2$) fragment peaks in the MS analysis (Fig. 3e, $K_0^{-1} = 2$ and 1.5). Crystal structures also revealed the anion packing and the pyridine stacking that stabilise each topological framework (Supplementary Figs. 6–8).

## Discussion

In summary, we have succeeded in the first selective synthesis of two polyhedral links of highly entangled, metal-linked peptide catenanes with the same ring and crossing numbers. A class of entangled compounds with high complexity has emerged through our folding-and-assembly strategy. There are great advantages to the rapidly developing DNA nanotechnology method[22,23], which can be used to manipulate a variety of advanced entangled nanostructures, but this method also has disadvantages, including its poor efficiency in three-dimensional shape construction, structural modification, and large-scale synthesis. However, these problems can be addressed by our folding-and-assembly method[16,20,24–26] if the structural prediction is accompanied hereafter. Considering the abundance of highly entangled nanostructures in biological systems and their contribution to the formation and stabilisation of giant protein assemblies[27–30], molecular entanglements can be considered a new nanoscale bonding pattern representing the principles of nature. This research thus brings us closer to the essential meaning and significance of mechanical bonds.

## Methods

**General information**. Boc-protected amino acids, 4 N HCl solution (in 1,4-dioxane), and coupling reagents and additives, 1-ethyl-3-(3-dimethylaminopropyl) carbodiimide hydrochloride (EDCI•HCl), 1-hydroxyl-1$H$-benzotriazole monohydrate (HOBt•H$_2$O), and $N,N$-diisopropylethylamine (DIEA) were purchased from Watanabe Chemical Industries. 3-Aminopyridine, nicotinic acid, AgBF$_4$, AgTf$_2$N, and AgOTf were purchased from TCI. AgPF$_6$ was purchased from Sigma-Aldrich. All chemicals were of reagent grade and used without any further purification. All NMR spectra were recorded on a Bruker Avance 500 MHz spectrometer equipped with a CP-TCI cryoprobe or a Bruker Avance III HD 500 MHz spectrometer equipped with a PABBO probe. ESI-MS data were recorded on a Waters micromass ZQ 2000 spectrometer, high-resolution MS (ESI-MS) data were recorded on a Bruker maXis spectrometer, and ion mobility MS data were recorded on a Bruker timsTOF Pro instrument. Preparative size-exclusion chromatography (SEC) was carried out using a JAIGEL W252 column (eluent: methanol or aqueous methanol). Melting points were determined on an MPA100 OptiMelt melting point apparatus (Stanford Research Systems). Elemental analyses were performed at the elemental analysis centre in the School of Science at the University of Tokyo.

**Synthesis of ligands**. Tripeptide ligands **3**, **5**, **7**, and **9** were synthesised using a route analogous to that published previously[25]. Detailed procedures and characterisation data of all compounds are shown in Supplementary Methods and Supplementary Figs. 30–47.

**Formation of [4]$_{12}$-catenane (4)**. CD$_3$NO$_2$ solutions (250 μL) of ligand **3** (25 mmol, 100 mM) and AgOTf (25 mmol, 100 mM) were prepared separately in micro vials. The two solutions were mixed and stirred for 1 min using a vortex mixer. On mixing, the solution immediately turned cloudy, and then gradually became clear over a longer time. Complete conversion to [4]$_{12}$-catenane **4** took 3 days; this process was monitored by $^1$H NMR measurements (see Supplementary Fig. 13). The initial concentration of the Ag(I) ion and its counter anion were optimised for the formation of **4** (Supplementary Figs. 14 and 15), and the stability of **4** with regard to temperature and dilution was also confirmed (see Supplementary Fig. 16). For comparison, conditions of a $T_2$-type [4]$_{12}$-catenane formation was also optimised using ligand **7** (Supplementary Figs. 17–19). NMR characterisation data of all [4]$_{12}$-catenanes are shown in Supplementary Figs. 20–29.

**Crystallographic analysis**. A single crystal of **4** was prepared by the vapour diffusion method. The CH$_3$NO$_2$ solution of **4** (100 μL, 20 mM) was placed in a capped microtube ($\varphi$ = 6 mm) with a tiny hole in the cap. Et$_2$O vapour was then slowly diffused into the microtube over 2 weeks at 20 °C, and colourless plate crystals were obtained. A single crystal of the [4]$_{12}$-catenane structure obtained from ligand **5** was also obtained with PF$_6^-$ counter anion using the same method. From ligand **7**, a single crystal of the [4]$_{12}$-catenane structure was obtained with BF$_4^-$ counter anion by slow concentration of the CH$_3$NO$_2$ solution. The diffraction data were collected on a Bruker APEX–II/CCD diffractometer equipped with a focusing mirror (Mo Kα radiation $\lambda$ = 0.71073 Å) and a cryostat system equipped with an N$_2$ generator (Japan Thermal Eng.) for all crystals. The crystals were removed from the solution, quickly attached to a loop of nylon fibre with antifreeze reagent (fluorolube, Sigma-Aldrich), and mounted on a goniometer. Data collection was performed at 100–108 K. The structures were solved by direct methods (SHELXS 2013/1)[31] and refined by full-matrix least-squares calculations on $F^2$ (SHELXL 2014/7)[32] using the SHELX-TL package. Hydrogen atoms were fixed at calculated positions and refined using a riding model. The structure of **6** was refined as a twin; this was detected using the PLATON/TwinRotMat tool[33]. Detailed crystallographic data are summarised in Supplementary Table 1. PyMOL 2.0 (Schrödinger, LLC) was used for the production of graphics.

## Data availability

The authors declare that the data supporting the findings of this study are available within the Supplementary Information files and from the corresponding authors upon reasonable request. The X-ray crystallographic coordinates for structures reported in this study have been deposited at the Cambridge Crystallographic Data Centre (CCDC), under deposition numbers 1869041–1869043. These data can be obtained free of charge from The Cambridge Crystallographic Data Centre via www.ccdc.cam.ac.uk/data_request/cif.

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

## Acknowledgements

This research was in part supported by Grants-in-Aid for Specially Promoted Research (24000009) for M.F. and Young Scientists (A) (JP15H05481) for T.S, and

Grants-in-Aid for Scientific Research on Innovative Areas (17H06460 and 17H06463) for K.S. We thank Dr. Shigeru Sakamoto, Dr. Ryo Kajita, and Yoshifumi Mori (Bruker Japan K.K.) for the ion mobility MS analysis.

## Author contributions

T.S. designed the project. T.S. and M.F. supervised the project and wrote the manuscript. A.S. and K.T. performed the experimental work. T.S., A.S. and M.F. analysed the results. T.S. and A.S. performed the crystallographic analysis. T.S., A.S., K.T. and Y.H. worked on the preliminary topological discussion, which was confirmed by K.S.

## Additional information

**Competing interests:** The authors declare no competing interests.

