## [Peer Review File · Nature Communications]

Reviewers' comments:

Reviewer #1 (Remarks to the Author):

This manuscript reports some outstanding results on interlocking ring structures.

The results are clearly reported and well illustrated. I have few remarks that need to be addressed before acceptance.

=====

There is some confusion in the way the compounds are called

we have

tripeptide 1 -P-G-P-

tripeptide 3 -T-P-P-

then

[4]12-catenane 2 for [Ag12(1)12](BF4)12

[4]12-catenane 4 for [Ag12(3)12](OTf)12

tripeptide 5 -A-P-P- gives

[5.Ag]12 on extended table 1 why not [Ag12(5)12](PF6)12

tripeptide 6 -I-P-P- gives

[6.Ag]12 on extended table 1 why not [Ag12(6)12](BF4)12

tripeptide 7 -V-P-P- gives

[7.AgBF4]12 on extended Data Figure 3 why not [Ag12(7)12](BF4)12

PLEASE make the nomenclature uniform:

Call all the compounds [Ag12(X)12][anion]12 and drop the extra numbers.

Assign the numbers at the tripeptides X=1,2,3,4,5 and then the compounds will be

[Ag12(X)12][anion]12

It would be helpful to update table 1 adding also the tripeptide -P-G-P- with [4]12-catenane 2 so there will be 3 examples of T2-tetrahedral link and 2 of three-crossed tetrahedral link

=====

The only evidence for the structure of [7.AgBF4]12 is with NMR? (extended data figure 3h).

Are the NMR data enough to proof the link topology? There are no single crystal data?

=====

About Figure 1 diagram. Where the diagrams come from?

Please use ref. 7 <http://knotilus.math.uwo.ca> that is able to draw all the 100 [4]12 catenane and identify the two links here reported giving the appropriate Gauss Code

http://katlas.org/wiki/Gauss_Codes or Dowker code [http://katlas.org/wiki/DT_\(Dowker-Thistlethwaite\)_Codes](http://katlas.org/wiki/DT_(Dowker-Thistlethwaite)_Codes)

Once you identify the link in knotilus, you will get a drawing that should be better than the one

reported where the crossings are too crowded.

Report the Gauss/Dowker code in the supp. material together with the archive number in knotilus, so anyone would be able to extract information about the two links here reported.

=====

References 10,12,13,14 are not about 12-crossing peptide [4]catenane as we may think reading the sentence on the first page :

Recently, we reported the folding and assembly(refs 10–14) of 12-crossing peptide [4]catenane 2 ($n = 4$, $c = 12$; this framework is hereafter abbreviated as [4]12-catenane) from tripeptide ligand 1 and Ag(I) ions (Scheme 1a).(ref 11)

Only ref 11 is pertinent. Ref 10,12,13,14 are about folding and assembly... but not of a 12-crossing 4-link!! Please move the four references where is more appropriate , or remove them.

Reviewer #2 (Remarks to the Author):

I'm a big fan of the Fujita group's work in general, and their reports on peptide metallo-catenanes are no exception to this. The work is well done and an important contribution to the field. However, I find the way the authors discuss the work rather disappointing. They highlight some things that are unimportant (e.g. the number of unique ways 4 rings can be interlocked) and ignore other things that are important (the assignment of the topologies in standard link tables and previously reported complex link topologies). I think the manuscript would be improved considerably by addressing these issues.

1. The stressing of the significance of having the same ring and crossing number for two different catenanes (in the title and the text). The authors make a big deal of the fact that they form two different links with the same number of rings (4) and crossing number (12). They also highlight that there are 100 different unique ways of connecting four rings (Figure 1a). Neither of these points are significant nor important. It's the equivalent of stressing that there are >250 different diastereoisomers possible with 8 asymmetric carbon atoms. By listing the number of 4 ring, 12 crossing link diastereomers in a Figure they give improper significance to this aspect to the casual reader.

2. On the other hand the authors completely ignore, and do not cite, other examples of complex molecular links. This fails to put the present manuscript in its proper context. Examples of complex molecular links that should, in my view, be cited include:
[Solomon link; 4-2-1 link] J. Am. Chem. Soc. 116, 375 (1994); J. Am. Chem. Soc. 121, 11014 (1999).
J. Am. Chem. Soc. 131, 920 (2009). Angew. Chem. Int. Ed. 52, 6464 (2013). Angew. Chem. Int. Ed. 53, 11261 (2014). Angew. Chem. Int. Ed. 54, 7555 (2015).
[6-3-1 link] Nat. Chem. 6, 978 (2014).
[6-3-2 link] Angew. Chem. Int. Ed. 57, 13833 (2018).
[9-3-7 link] Nat. Chem. 10, 1083 (2018).

3. Page 1, abstract: "we report the selective construction of two topologies of 12-crossing peptide [4]catenanes ($n = 4$, $c = 12$), which have the largest crossing number among all known interlocking molecules". Not true. There are interlocking protein chainmail networks [Science 289, 2129 (2000)], synthetic catenane polymers [Science 358, 1434 (2017)] and DNA links which have larger crossing

numbers.

4. Page 2: "We thus suggest that concerted folding and assembly may be a rational strategy for the generation of highly complex polyhedral link structures with $c > 10$, which, until now, have only been addressed by the DNA nanotechnology method."

The authors did not predict the way that the metallo-peptides were going to fold to form the different links, unlike DNA nanotechnology, and so it is incorrect and misleading to suggest that this method can currently be used in predictive design.

5. Page 8: "However, these problems can be addressed by our new method, which can be described as the peptide origami method."

As noted above this is rather nonsense. The assembly and folding method is, as yet, nowhere near being predictable in terms of what structures will be formed. It is not a peptide form of DNA origami.

6. The authors should state which link their compound corresponds to in the standard link tables, i.e. the Rolfsen link table and the Thistlethwaite link table [Menasco, W. & Thistlethwaite, M. The classification of alternating links. *Ann. Math.* 138, 113–171 (1993)]. That categorises which link has been made from a topological standpoint.

Reviewer #3 (Remarks to the Author):

The manuscript reports new interesting achievements in the construction under control of entangled complex systems. Here the authors describe the selective synthesis of two topological isomers of 12-crossing peptide [4]catenanes (the tetrahedral link and the three-crossed tetrahedral link) by changing the sequence of the tripeptide. The selective assembly of one of the two topological [4]catenanes is unambiguously related to the different conformation of the tripeptide sequence. The present results are very important in the view of developing artificial complex systems under control. My recommendation is to publish with minor revisions. In the following more detailed comments.

I suggest to add in Table 1 the information relative to the conformation adopted by the tripeptide to remark its influence on the final topological type adopted by the catenane.

The adopted nomenclature is somehow confusing. It seems that even numbers refer to catenanes while odd numbers refer to tripeptides, however, tripeptides I-P-P and V-P-P are named 6 and 7, respectively, while no numbers are assigned to their corresponding catenanes. For the sake of clarity, the adopted numbering scheme should be used for all described species.

No comments are made in the main text on the interactions that stabilize the whole assemblies, apart the formation of hydrogen bonded trimers involving the OH groups in the T-P-P tripeptide based three-crossed catenane. Even if such interactions are described in the supporting material a short comment in the main text could be added.

Figure 1b of the manuscript: The meaning of the arrows on the four rings of the two topological links is not clear and it seems to me that is not commented within the text. An explanation should be helpful for the reader.

Concerning the relationship between three-crossed and cuboctahedral links: I suggest to modify last sentence at pag 6: "Figure 3b,c clearly illustrates that the three-crossed tetrahedral link and the cuboctahedral link can be topologically transformed into the other." Adding at the end "...without the

need to cut chemical bonds." to make more clear to the reader the relationship between these two links. Is this a pure topological speculation or isomerization in solution are possible? In general, are isomerization between different links possible in solution? Can NMR technique be able to distinguish different topological links for the [4]12-catenane?

The evidence for adopting one of the two links comes from the SCXR analysis. In the case of the catenane obtained with ligand 7 the topology was assigned as T2 but the crystal structure was not determined. How was assigned this link topology? It is not clear to me if from NMR data it is possible to distinguish different link topology. Can the Author comment on this aspect?

All the [4]12-catenane reported here have been assembled in nitromethane but previous examples were assembled in different solvents. Is there any effect of the solvent on the folding and assembly process?

Responses to Reviewers

For reviewer 1:

> This manuscript reports some outstanding results on interlocking ring structures. The results are clearly reported and well illustrated.

We appreciate the reviewer's very positive evaluation.

> There is some confusion in the way the compounds are called...

PLEASE make the nomenclature uniform:

Call all the compounds [Ag12(X)12][anion]12 and drop the extra numbers.

Assign the numbers at the tripeptides X=1,2,3,4,5 and then the compounds will be [Ag12(X)12][anion]12

According to this suggestion, we revised the compound numbers in the revised text and SI. Since we need compound numbers for each [4]₁₂-catenanes, we re-numbered as follows:

Odd numbers: peptide ligands

Even numbers: [4]₁₂-catenanes

We described the counter anion information, e.g. [4]₁₂-catenane **6** (counter anion: OTf⁻), if necessary. Due to the addition of compound numbers for [4]₁₂-catenane **6**, **8** and **10**, we newly added two sentences below for the clearer explanation of their topologies in the revised text:

*“In the same manner, the topology of **8** and **10** also belongs to T₂-tetrahedral link.”*

*“The topology of **6**, is the same as that of **4**.”*

> It would be helpful to update table 1 adding also the tripeptide -P-G-P- with [4]₁₂-catenane 2 so there will be 3 examples of T₂-tetrahedral link and 2 of three-crossed tetrahedral link

According to this suggestion, we put the tripeptide -P-G-P- in the table 1.

> The only evidence for the structure of [7.AgBF₄]₁₂ is with NMR? (extended data figure 3h). Are the NMR data enough to proof the link topology? There are no single crystal data?

Preparation of single crystals and X-ray analyses requires a lot of efforts as well as luck. Unfortunately, we could not obtain good quality of crystals for the V-P-P ligand, but we believe the identical ¹H NMR pattern shown in Extended Fig. 3, f and g clearly confirms the same framework topology.

> About Figure 1 diagram. Where the diagrams come from?

Please use ref. 7 <http://knotilus.math.uwo.ca> that is able to draw all the 100 [4]₁₂ catenane and identify the two links here reported giving the appropriate Gauss Code http://katlas.org/wiki/Gauss_Codes or Dowker code [http://katlas.org/wiki/DT_\(Dowker-Thistlethwaite\)_Codes](http://katlas.org/wiki/DT_(Dowker-Thistlethwaite)_Codes)

Once you identify the link in knotilus, you will get a drawing that should be better than the one reported where the crossings are too crowded.

Report the Gauss/Dowker code in the supp. material together with the archive number in knotilus, so anyone would be able to extract information about the two links here reported.

According to this suggestion, we improved the Fig. 1 diagrams in the revised manuscript. We also put the knotilus archive codes (12x-4-13 for compound **2**, 12x-4-92 for compound **4**) in the revised

SI. All the readers can easily access to the other codes such as the Gauss/Dowker code. We appreciate this valuable suggestion.

> *References 10,12,13,14 are not about 12-crossing peptide [4]catenane as we may think reading the sentence on the first page : ...*

Only ref 11 is pertinent. Ref 10,12,13,14 are about folding and assembly.... but not of a 12-crossing 4-link!! Please move the four references where is more appropriate , or remove them.

We removed the citations 10, 12–14 here in the revised text.

For reviewer 2:

> *I'm a big fan of the Fujita group's work in general, and their reports on peptide metallo-catenanes are no exception to this. The work is well done and an important contribution to the field.*

We appreciate the reviewer's very positive evaluation.

> *They highlight some things that are unimportant (e.g. the number of unique ways 4 rings can be interlocked) and ignore other things that are important (the assignment of the topologies in standard link tables and previously reported complex link topologies).*

First, we don't ignore the previous reported chemical topologies. We have cited the critical reviews (refs 2 and 3) and the comprehensive book (ref 4), which cover all the complex link topologies reported. Second, the complex topologies reported in this work cannot be described by the previous standard descriptions because of the large crossing number 12: The Alexander–Briggs notation only covers up to 9 crossings, the Rolfsen table do up to 10 crossings, and the Thistlethwaite link table do up to 11 crossings (see the URL below). Thus, we found that the descriptions as polyhedral links were reasonable here. To clearly describe this situation, we put the sentence, "In case of > 11-crossings, the existence of very large number of topologies prevents us from assigning the topology by standard link tables as seen in previous topological molecules,²⁻⁴ and accordingly we choose to use the polyhedral link description here:" in the revised text. Therefore, we emphasize that the chemical topology field is entering a new phase by emergence of 12-crossing [4]catenanes that cannot be described by previous standard methods.

Please also see the website on link tables:

http://katlas.org/wiki/Main_Page

> *1. The stressing of the significance of having the same ring and crossing number for two different catenanes (in the title and the text). The authors make a big deal of the fact that they form two different links with the same number of rings (4) and crossing number (12). They also highlight that there are 100 different unique ways of connecting four rings (Figure 1a). Neither of these points are significant nor important. It's the equivalent of stressing that there are >250 different diastereoisomers possible with 8 asymmetric carbon atoms. By listing the number of 4 ring, 12 crossing link diastereomers in a Figure they give improper significance to this aspect to the casual reader.*

Answer to this comment is related to the one before. In the field of the chemical topology, no one encountered the problem (as well as interests) of the exponential increase of the number of topologies at large crossing numbers to date. As obtaining two types of 12-crossing [4]catenane structures, we first recognized that researchers in the chemical topology fields needed to begin considering how to construct one topology selectively among large number of existing topologies after this. Likewise, the previous standard topological descriptions become no longer practical (and

impossible in this work) for large number of existing topologies at the same ring- and crossing-numbers. We chose the classification of polyhedral links for better understanding of two topologies to readers here. Therefore, we consider it is the significant issue that very large number of topologies exists even at the same ring- and crossing-numbers.

> *On the other hand the authors completely ignore, and do not cite, other examples of complex molecular links. This fails to put the present manuscript in its proper context. Examples of complex molecular links that should, in my view, be cited include:*

[Solomon link; 4-2-1 link] J. Am. Chem. Soc. 116, 375 (1994); J. Am. Chem. Soc. 121, 11014 (1999). J. Am. Chem. Soc. 131, 920 (2009). Angew. Chem. Int. Ed. 52, 6464 (2013). Angew. Chem. Int. Ed. 53, 11261 (2014). Angew. Chem. Int. Ed. 54, 7555 (2015).

[6-3-1 link] Nat. Chem. 6, 978 (2014).

[6-3-2 link] Angew. Chem. Int. Ed. 57, 13833 (2018).

[9-3-7 link] Nat. Chem. 10, 1083 (2018).

Again, we would like to explain that we have not ignored these works. These works are seen in the refs 2–4 except for the last two (published quite recently). However, according to this comment, we added new citations to highlight earlier topological molecules in the revised text (refs R1–R8). We're afraid excessive citations confuse the readers, therefore we selected the first one or two, well-characterized examples of prime links:

R1–R2, Solomon link: *J. Am. Chem. Soc.* **116**, 375 (1994) and *J. Am. Chem. Soc.* **121**, 11014 (1999)

R3, Borromean rings: *Science* **304**, 1308 (2004)

R4, Star of David: *Nature Chem.* **6**, 978 (2014)

R5–6, 6-3-3 link: *Nature Chem.* **7**, 354 (2015) and *Angew. Chem. Int. Ed.* **54**, 2796 (2015)

R7, 6-2-3 link: *Angew. Chem. Int. Ed.* **130**, 14029 (2018).

R8, 9-3-7 link: *Nature Chem.* **10**, 1083 (2018)

> *3. Page 1, abstract: “we report the selective construction of two topologies of 12-crossing peptide [4]catenanes ($n = 4$, $c = 12$), which have the largest crossing number among all known interlocking molecules”. Not true. There are interlocking protein chainmail networks [*Science* 289, 2129 (2000)], synthetic catenane polymers [*Science* 358, 1434 (2017)] and DNA links which have larger crossing numbers.*

We have intended to describe the 12-crossings is the largest among the synthetic discrete molecules to date. Therefore, for clarity, we replaced the term “among all known interlocking molecules” with “among all synthetic well-defined interlocking molecules to date” in the revised text.

> *4. Page 2: “We thus suggest that concerted folding and assembly may be a rational strategy for the generation of highly complex polyhedral link structures with $c > 10$, which, until now, have only been addressed by the DNA nanotechnology method.”*

The authors did not predict the way that the metallo-peptides were going to fold to form the different links, unlike DNA nanotechnology, and so it is incorrect and misleading to suggest that this method can currently be used in predictive design.

According to this suggestion, we removed the latter part of text, “, which, until now, have only been addressed by the DNA nanotechnology method” in the revised text.

> *5. Page 8: “However, these problems can be addressed by our new method, which can be described as the peptide origami method.”*

As noted above this is rather nonsense. The assembly and folding method is, as yet, nowhere near being predictable in terms of what structures will be formed. It is not a peptide form of DNA origami.

According to this suggestion, we replaced the sentence with “*However, these problems can be addressed by our folding-and-assembly method if the structural prediction is accompanied hereafter.*” in the revised text.

> *6. The authors should state which link their compound corresponds to in the standard link tables, i.e. the Rolfsen link table and the Thistlethwaite link table [Menasco, W. & Thistlethwaite, M. The classification of alternating links. Ann. Math. 138, 113–171 (1993)]. That categorises which link has been made from a topological standpoint.*

The answer to this comment is the same as the second and third comments of this reviewer.

For reviewer 3:

> *The present results are very important in the view of developing artificial complex systems under control.*

We appreciate the reviewer’s very positive evaluation.

> *I suggest to add in Table 1 the information relative to the conformation adopted by the tripeptide to remark its influence on the final topological type adopted by the catenane.*

The adopted nomenclature is somehow confusing. It seems that even numbers refer to catenanes while odd numbers refer to tripeptides, however, tripeptides I-P-P and V-P-P are named 6 and 7, respectively, while no numbers are assigned to their corresponding catenanes. For the sake of clarity, the adopted numbering scheme should be used for all described species.

According to this suggestion, we put the peptide conformation in the revised table 1. We also rearranged the compound numbers. (This suggestion is similar to the comment by reviewer 1.)

> *No comments are made in the main text on the interactions that stabilize the whole assemblies, apart the formation of hydrogen bonded trimers involving the OH groups in the T-P-P tripeptide based three-crossed catenane. Even if such interactions are described in the supporting material a short comment in the main text could be added.*

According to this suggestion, we put the sentence “*Crystal structures also revealed the anion packings and pyridine stackings that stabilize each topological framework (Figs. S5–S7).*” in the revised text.

> *Figure 1b of the manuscript: The meaning of the arrows on the four rings of the two topological links is not clear and it seems to me that is not commented within the text. An explanation should be helpful for the reader.*

According to this suggestion, we put the sentence “*Each ring has a directional sense, which is defined by the N→C direction within each peptide ligand here.*” at the end of the Fig. 1 caption in the revised text.

> *Concerning the relationship between three-crossed and cuboctahedral links: I suggest to modify last sentence at pag 6: “Figure 3b,c clearly illustrates that the three-crossed tetrahedral link and the cuboctahedral link can be topologically transformed into the other.” Adding at the end “....without the need to cut chemical bonds.” to make more clear to the reader the relationship between these two links.*

We appreciate the reviewer’s suggestion and questions. This paragraph describes the pure

topological discussion not about the molecular structure. Therefore, we put the text “*without the need to cut any loops.*” in the revised text.

> Is this a pure topological speculation or isomerization in solution are possible? In general, are isomerization between different links possible in solution? Can NMR technique be able to distinguish different topological links for the [4]₁₂-catenane?

We appreciate the reviewer’s suggestion and questions. We have analyzed possibility of the solution-state isomerization by changing temperature and concentration of complexes at the ¹H NMR measurement as shown in supplementary materials. In order to link the text and these data, we put the text below in the revised text.

*“Variable temperature NMR analysis reveals that the topological interconversion between three-crossed type and T₂-type [4]₁₂-catenane does not occur in solution from 273 K 353 K (Figs. S11a and S14). Similarly, the concentration change before and after the complexation also does not affect the topological selectivity (Figs. S11b and S12) except for the formation of the stable intermediates at the low concentration in case of **4** (Fig. S9).”*

> The evidence for adopting one of the two links comes from the SCXR analysis. In the case of the catenane obtained with ligand 7 the topology was assigned as T2 but the crystal structure was not determined. How was assigned this link topology? It is not clear to me if from NMR data it is possible to distinguish different link topology. Can the Author comment on this aspect?

According to this suggestion, we put the sentence “*in ¹H NMR observation: the T₂-type [4]₁₂-catenane (**10**) was formed*” in the revised text.

> All the [4]₁₂-catenane reported here have been assembled in nitromethane but previous examples were assembled in different solvents. Is there any effect of the solvent on the folding and assembly process?

All the [4]₁₂-catenanes including the previous report (ref 11) are self-assembled basically only in nitromethane. (In the previous report, we also used MeCN for the self-assembly, but this is not the same metal (not silver(I) but gold(I)). The exploration on various solvent effects for the folding and assembly process can be an interesting topic but requires a lot of experiments, therefore we would like to investigate it as a future publication.

REVIEWERS' COMMENTS:

Reviewer #1 (Remarks to the Author):

The authors fully reply to my comments
The paper can now be accepted without any further changes.

Reviewer #2 (Remarks to the Author):

I was not aware that published link tables were limited to 11 crossings. I'm content that the authors have at least considered my suggestions for improving their manuscript and the revised manuscript is OK as far as I'm concerned.

Reviewer #3 (Remarks to the Author):

The Authors have taken into account all the points arose in the previous referee process and revised the manuscript accordingly. My opinion is that the manuscript can be accepted in its present form.